# Resilience Improves the Quality of Life and Subjective Happiness of Physiotherapists during the COVID-19 Pandemic

**DOI:** 10.3390/ijerph19148720

**Published:** 2022-07-18

**Authors:** Patricia Angeli da Silva Pigati, Renato Fraga Righetti, Victor Zuniga Dourado, Bruna Tiemi Cunha Nisiaymamoto, Beatriz Mangueira Saraiva-Romanholo, Iolanda de Fátima Lopes Calvo Tibério

**Affiliations:** 1Faculdade de Medicina FMUSP, Universidade de São Paulo, São Paulo 02146-903, SP, Brazil; refragar@gmail.com (R.F.R.); bruna.tiemi@fm.usp.br (B.T.C.N.); beatriz.msaraiva@fm.usp.br (B.M.S.-R.); iocalvo@uol.com.br (I.d.F.L.C.T.); 2Hospital Sírio-Libanês, São Paulo 01308-050, SP, Brazil; 3Department of Human Movement Sciences, Federal University of São Paulo, Santos 11015-020, SP, Brazil; victor.dourado@unifesp.br; 4Public Employee of São Paulo (IAMSPE), São Paulo 04029-000, SP, Brazil

**Keywords:** physiotherapists, COVID-19, resilience, quality of life, happiness

## Abstract

Resilience is an individual characteristic that protects mental health. However, its impact on the lives of Brazilian physiotherapists during COVID-19 is not known. This study aimed to analyze whether resilience modulates the perceived quality of life (QoL) and subjective happiness (SH) of physiotherapists who work with COVID-19 patients, compared with those who do not. A cross-sectional study was conducted between 22 August and 22 October 2020. Physiotherapists working in critical and non-critical hospital sectors were invited to participate in the study. The participants completed sociodemographic questionnaires and were graded on the 14-item Resilience Scale, 36-item Short-Form Health Survey (SF-36), and the Subjective Happiness Scale. In total, 519 physiotherapists were enrolled in the study. Physiotherapists with low resilience who worked with COVID-19 patients reported lower scores on the SF-36 subscales (except for social functioning) and the Subjective Happiness Scale, compared with those with high resilience who did not work with COVID-19 patients. These responses were modulated by age, sex, absence from work, receipt of personal protective equipment, host leadership, and practice and maintenance of regular physical activity. In conclusion, physiotherapists with low resilience who worked with COVID-19 patients presented lower perceptions of QoL and SH, compared with the other study participants.

## 1. Introduction

According to the World Health Organization (WHO), Brazil has the third-highest number of cases and the second-highest number of deaths from the coronavirus disease 2019 (COVID-19) worldwide [1]. However, the government of Brazil—a populous country of approximately 213,364,448 million people [2]—has not adopted strict containment policies, resulting in over 660,000 deaths to date [3,4].

The first case of COVID-19 in Latin America was notified in the state of São Paulo, a pandemic epicenter in the country [5,6]. It is the most populated state of Brazil, with 645 counties and 46,649,132 million people [7]. As elsewhere in the world, hospitalization of severe COVID-19 cases overburdened health services and frontline healthcare workers (FHCWs) in the state [8,9,10,11].

From the perspective of hospitals, there was a need to increase intensive care beds, mechanical ventilators [12], personal protective equipment (PPE) [13], and hire trained professionals to manage critically ill patients [14]. It was especially important to segregate COVID-19 and non-COVID-19 patients [15]. Additionally, it was important to safeguard the mental integrity of FHCWs who were exposed to numerous physical and psychological stressors [16]. Serious negative mental health consequences have been observed in FHCWs who provided direct assistance to COVID-19 patients during the acute phase of the disease [17,18,19]. This population is constantly exposed to infections and other stressful circumstances, such as fear of contagion and infecting others, social distancing, inadequate support and inadequate preparation, while working under the pressure of critical patient management [20].

Physiotherapists are frontline professionals essential for the management of critically ill patients. Physiotherapists in COVID-19 isolation hospitals have close contact with patients during aerosols generating activities such as exercise, and early mobilization, and pulmonary procedures [21]. They also handle oxygen administration, invasive and non-invasive mechanical ventilation, and capacity assessment during hospital admission and discharge [22]. The number of physiotherapists needed per hospital bed determined by the Federal Council of Physiotherapy and Occupational Therapy is based on the patient’s severity profile. During a six-hour shift, one physiotherapist is recommended for every six to ten patients in ICU, semi-intensive, urgent, and emergency conditions; one for eight to ten patients in wards; and one for every twelve ambulatory patients [23]. During the pandemic, an increase in the number of ICU beds spiked the workload of frontline professionals [24], exposing them to physical and mental stressors. Consequently, symptoms of burnout and emotional exhaustion [25], stress [26], and poor sleep quality [27] have been reported among these professionals during the COVID-19 pandemic.

Experience with other pandemics, such as severe acute respiratory syndrome (SARS) and Ebola, shows that high levels of psychological distress may decrease the quality of life (QoL) of FHCWs [28,29]. Poor perceived QoL is also reported by doctors, nurses, and healthcare assistants working with COVID-19 patients [30]. Overwhelming workload, inadequate knowledge and training, and fear of infection are a few stress factors associated with symptoms of depression, anxiety, and poor QoL [31]. In this way, in the aspect of viral pandemics, FHCWs have their psychological well-being threatened.

The subjective happiness (SH), considered as a direct indicator of QoL [32], has a direct relationship with mental health [33]. Happiness, or well-being, is a positive emotional state associated with a perception of life satisfaction and depends on the interaction of genetic, emotional, and cognitive factors [34,35,36]. Furthermore, unpredictable events and circumstances can change one’s perception of happiness. A study in a hospital in India during the pandemic revealed that perceived stress decreases SH in healthcare workers [37]. Stress, unhappiness, and phycological suffering at work increase absenteeism rates, risk of accidents, and absences due to physical and psychological illnesses [38,39]. In this sense, balanced work environments align quality and profitability with QoL, well-being, and worker happiness [40]. Although the terms happiness or well-being and QoL are often used synonymously, they are different constructs but complementary and closely related. While QoL focuses more on physical health and objective issues involving human beings, happiness is characterized as a psychological state or feeling of contentment with life [41]. Both concepts can be influenced by human strength knowledge as resilience.

Resilience has been described as a protective factor that can minimize the negative impact of difficult situations, prevent psychological disorders [42], help manage stress and unhappiness, and enable recovery. Different groups of researchers have explored the relationship between high levels of resilience and better QoL. These studies were performed with people in challenging circumstances such as patients with cancer [43], in old age [44], in medical students [45], war [46] and disaster survivors [47], and in the job [48].

The resilience role as an important personal resource to inhibit burnout was investigated by Ferreira & Gomes (2021) in the period of the second wave of COVID-19 in Portugal. The research findings indicated that resilience could protect healthcare professionals against the negative consequences of job strain [49]. The combination of being a healthcare worker and a resilient person has proved to be useful when facing adverse professional situations [50]. The possibility of mediating the effect of resilience on the QoL of healthcare workers was described by Son et al. (2022). The authors found that better QoL was predicted by a lower level of anxiety response to the viral epidemic, a lack of current psychiatric symptoms, a higher level of resilience, and higher organizational commitment [51]. Choi et al. (2022) showed that low resilience adversely affects the QoL and mental health of FHCWs, which directly interferes with the quality of services provided to patients [52].

A subjective sense of happiness may be increased by resilience, reducing the negative physiological effects [53]. In a study with university students, Lü et al. (2022) revealed that resilience was a potential mediator in the relationships among extraversion, neuroticism. and happiness and might be a key factor in enhancing happiness and positive affect, which serves to improve well-being [54].

Thus, resilience is indispensable for mental and general health, well-being, and high QoL [55]. Although highly resilient people generally face adversities with equanimity and a sense of control over their environment [56], several factors may influence this positive response, such as personal attributes, social factors, and coping conditions.

Strategies for handling adversities and maintaining well-being, as described by Haglund et al. (2007), include: acknowledging and accepting problems, facing fears, seeking social support, adaptability, optimism, reframing stressors in a positive light, and performing physical exercise [57]. This knowledge is important for designing public health policies to prioritize the mental health of FHCWs. Since the beginning of the COVID-19 pandemic, researchers have been emphasizing actions that improve resilience in FHCWs [58,59].

Resilience is a personal attribute that helps healthcare professionals face adverse situations at work and improves their professional QoL. Maintenance of physiological well-being can protect against stressful events, minimizing psychological sequelae and increasing the ability to address emergencies effectively [60]. Positive emotions and attitudes, optimism, and a sense of humor are characteristics of resilient individuals [61].

The hypothesis of the present study was whether resilience could modulate SH and QoL in Brazilian physiotherapists. Therefore, we investigated whether resilience modulates the perception of QoL and SH of physiotherapists who work with COVID-19 patients, compared with those who do not, in both critical areas (critical care units, semi-intensive units, and emergency rooms) and non-critical areas (wards, ambulatory services, and supervisory settings). Additionally, we explored the determinants factors for these responses.

## 2. Material and Methods

### 2.1. Study Design and Participants

A cross-sectional observational study was conducted with physiotherapists in hospitals in São Paulo, Brazil. Eligible participants included those who provided free and informed consent and worked in critical care units, semi-intensive units, wards, emergency rooms, supervisor occupations, and ambulatory. Only those participants who indicated on the form that they work in hospitals in São Paulo were given access to the questionnaire. They were also asked whether they work in sectors intended for COVID-19 patients. The participants could opt out of the survey at any time.

### 2.2. Data Collection and Ethical Considerations

The data were collected from 22 August to 22 October 2020, through an anonymous online questionnaire using Google Forms (Google LLC., Menlo Park, CA, USA). Participants were recruited via two non-probability sample methods: convenience sampling (involving members of hospitals accessible to the study team) and snowball sampling (recruiting respondents among acquaintances of participants). Questionnaires were distributed via e-mail and different social media platforms (WhatsApp (WhatsApp Inc., Menlo Park, CA, USA), Facebook (Facebook Inc., Menlo Park, CA, USA), Instagram (Menlo Park, CA, USA), given the difficulty in accessing the participants due to the high infection rate in São Paulo. This study was approved by the Ethics and Research Committee of the Hospital of Clinics of the Faculty of Medicine at the University of São Paulo (n° 4,229,228). All participants provided informed consent electronically before registration.

### 2.3. Demographic Data

Primary demographic data included information on age, sex, pregnancy status, marital status, whether the participant has children, lives with seniors, lives with children, and has experienced a death in the family or the death of a close friend due to COVID-19; graduation date; regular physical activity; maintenance of physical activity during the pandemic period; chronic disease history; absence from work due to other diseases; COVID-19 diagnosis; hospitalization due to COVID-19; the nature of the institution in which the participant works; removal from work; the hospital sector in which the participant works; weekly workload, wage/income, salary reduction during the pandemic period, receipt of PPE, receipt of host leadership and receipt of training.

### 2.4. 14-Item Resilience Scale (14-RS) 

The 14-item Resilience Scale (see Appendix A) was adapted and validated to Brazilian Portuguese by Pesce et al. (2005) [62]. The scale is designed for self-administration and uses a 7-point Likert scale for each item: 1 (strongly disagree) to 7 (strongly agree). A short form measure of resilience comprises 14 items clustered in five domains: self-reliance, meaning, equanimity, perseverance, and existential aloneness. The total score is categorized as follows: very low (14–56 points), low (57–64 points), moderately low (65–73 points), moderately high (74–81 points), high (82–90 points), and very high (91–98 points) [63]. We dichotomized the scale in two ways, using low resilience (14–73) and high resilience (74–98) as cut-offs. In the present study, Cronbach’s alpha for this scale was 0.918.

### 2.5. 36-item Short Form Health Survey (SF-36)

The 36-item Short Form Health Survey questionnaire was used to measure health-related QoL. This questionnaire comprises 36 questions on a Likert scale with eight subscales to evaluate the physical function, social functioning, role limitations due to physical problems, and role limitations due to emotional problems, mental health, vitality, pain, and general health perception (see Appendix B). The total score for each SF-36 subscale ranges between 0 and 100, where higher scores indicate better health [64]. An additional unscored item (question 2) compares the patient’s perception of his/her current health with that of the previous year; higher scores indicate worsening health perceptions [65]. The SF-36 was validated for use in the native language of the physiotherapists who participated in the study [66]. In the present study, Cronbach’s alpha for this questionnaire was 0.742.

### 2.6. Subjective Happiness Scale (SHS)

The Subjective Happiness Scale measures SH and includes a 4-item instrument with a 7-point Likert-type scale, where 7 indicates the happiest and 1 least happy (see Appendix C). Among its four items, item 1 is a general self-appraisal of the respondents’ lives; item 2 measures respondents’ relationships with others; and items 3 and 4 evaluate agreement with short statements characterizing happy and unhappy people [67]. The final score is the average of the four answers, and higher scores indicate higher happiness levels. The transcultural version in Portuguese was developed by Pais-Ribeiro in 2012 [68]. In the present study, the Cronbach’s alpha for this scale was 0.816.

### 2.7. Statistical Analysis

The data were initially checked descriptively. The normality of the variables was evaluated using the Kolmogorov–Smirnov test. Continuous variables are presented as medians (interquartile ranges) because of non-constant data distribution. Categorical variables are presented as frequencies (percentages). We used the Mann–Whitney test to compare the studied variables between the low- and high-resilience groups, and the COVID and NO COVID groups. Categorical variables were compared between these groups using the c^2^ test.

After analyzing the univariate associations, we identified several trends in the multiple linear regression results to determine the influence of resilience on the mental health outcomes of the health professionals studied. The scores of the SF-36 and SHS questionnaires determined the outcomes. Resilience as a dichotomous variable (i.e., low or high) and working with COVID-19 patients were considered the main predictor. After analysis, we adjusted all the trends for the following variables: age, sex, absence from work, receipt of protective equipment, receipt of host leadership, regular physical activity, and maintenance of physical activity during the pandemic period. We also investigated the possible interaction between resilience and working with COVID-19 patients for all the outcomes assessed. Multi-collinearities were prevented by considering a variation inflation factor < 4 between the predictors and co-variables. Calculations on the sample data were performed using www.statstodo.com (accessed on 7 October 2021). Based on the trends determined from the multiple linear regression analysis, we estimated that the sample needed to consider a conservative multiple R (i.e., the multiple regression effect size) equal to 0.20 and up to 10 predictors to be included in each model. Using alpha = 0.05 and beta = 0.20 (i.e., statistical power = 0.80), we identified that a sample of 398 participants would conclusively meet our research objectives. Thus, the sample size of this study is over 30.4% higher than the calculated required size. All analyses were performed using the Statistical Package SPSS version 24 (IBM Corp., Armonk, NY, USA), and the r alpha probability error was established at 5%.

## 3. Results

In total, 603 physiotherapists responded to the questionnaires, but 84 were excluded because they did not work in a hospital in São Paulo. Therefore, 519 participants completed the survey, who were categorized in terms of low (145 (38.9%)) and high (374 (61.1%)) resilience and working with COVID-19 patients (COVID group, 445 (72.7%)) or not (NO COVID group, 74 (27.3%)).

### 3.1. Demographic Characteristics of Groups Categorized in Terms of Resilience and Working with COVID-19 Patients

Table 1 shows the demographics, socioeconomic status, and clinic dates of participation in the present study for the low- and high-resilience groups. We found that the participants in the high-resilience group (374 (61.1%)) were more physically active, compared with to the low-resilience group (203 (54.3%) vs. 64 (44.1%)) and had more support for coping with the pandemic through receiving host leadership (263 (70.3%) vs. 83 (57.2%)) and training (307 (82.1%) vs. 107 (73.8%)) (*p* < 0.05).

Table 2 shows the same characteristics for the COVID and NO COVID groups, where 445 participants [72.7% of the sample] were in the COVID group. The NO COVID group, compared with the COVID group, had a higher percentage of participants who were pregnant (7 (9.5%) vs. 6 (1.3%)), were living with children (40 (54.1%) vs. 147 (33.0%)), had graduated between 11 and 20 years ago (38 (51.4%) vs. 160 (36.0%)) or between 21 and 30 years ago (9 (12.2%) vs. 20 (4.5%)), engaged in physical activity during the pandemic period (19 (25.7%) vs. 61 (13.7%)), and had a salary up to R$ 7.000,00 (13 (17.6%) vs. 35 (7.9%)). The NO COVID group also included participants who had received a large salary reduction. Most of the participants in the NO COVID group, compared with the COVID group (*p* < 0.05), worked in private hospitals (28 (37.8%) vs. 130 (29.2%)), wards (30 (40.5%) vs. 66 (14.8%)), ambulatory medicine (9 (12.2%) vs. 2 (0.4%)), and supervisor occupations (5 (6.8%) vs. 8 (1.8%)), and received more host leadership as support for coping with the pandemic (62 (83.8%) vs. 284 (63.8%)).

Significant features of the COVID group, compared with the NO COVID group (*p* < 0.05), were that the participants graduated between 5 and 10 years ago (159 (35.7%) vs. 14 (18.9%)), were absent from work because of other diseases (64 (14.4%) vs. 2 (2.7%)), mostly worked in critical care units (348 (78.2%) vs. 26 (35.1%)), and had a workload between 51 and 60 h per week (68 (15.3%) vs. 3 (4.1%)).

### 3.2. Physiotherapists’ Perceptions of Health (Question 2/SF-36)

The scores for the physiotherapists’ perceptions of their current health, compared with that of the previous year, were determined for the low- and high-resilience and COVID and NO COVID groups are demonstrated in Table 3.

### 3.3. SF-36 and SHS for Groups Categorized in Terms of Resilience and Working with COVID-19 Patients

The low-resilience group presented lower scores (median (IQR)) on the SF-36, compared with the high-resilience group: physical functioning (80 [60–90] vs. 90 [75–100]), role-physical (50 [25–100] vs. 75 [25–100]), bodily pain (51 [41–62] vs. 62 [51–84]), general health (60 [50–73] vs. 72 [60–82]), vitality (35 [20–45] vs. 50 [35–65]), role-emotional (33 [0–66] vs. 100 [91–100]), and mental health (44 [36–52] vs. 66 [0–100]), p < 0.001. There was no difference in social functioning between the low- and high-resilience groups (50 [37–50] vs. 50 [37–50]), *p* = 0.163. The low-resilience group presented lower scores for SH, compared with the high-resilience group (4.25 [3.62–5.00] vs. 5.50 [5.00–6.25]), *p* < 0.001 (Table 4).

Table 5 shows that participants working with COVID-19 patients had lower scores (median (IQR)) on the SF-36, compared with those who do not work with COVID-19 patients (p < 0.001): physical functioning (85 [70–95] vs. 95 [85–100]), role-physical (50 [25–100] vs. 100 [75–100]), bodily pain (51 [41–72] vs. 84 [62–100]), general health (67 [55–77] vs. 80 [61–90], vitality (40 [30–55] vs. 65 [55–80]), role-emotional (33 [0–100] vs. 100 [91–100]), and mental health (52 [44–64] vs. 82 [68–92]). There was no difference in social functioning between the COVID and NO COVID groups (50 [37–50] vs. 50 [37–50]), *p* = 0.638. The COVID group presented lower scores for SH, compared with the NO COVID group (5.00 [4.25–5.75] vs. 6.12 [5.25–6.75]), *p* < 0.001.

### 3.4. Regression Analysis

The results of the univariate analysis showed that unadjusted factors of resilience and working with COVID-19 patients were significant predictors of the SF-36 and SH. After performing a multivariate analysis, these variables remained significant predictors with similar coefficients of magnitude to the unadjusted values, showing the independence of these predictors. Resilience and working with COVID-19 patients significantly explained between 14.0% and 44.8% of the total variability in the areas covered by the SF-36 questionnaire and 25% of happiness. After multivariate adjustment, the remaining nine predictors added between 0.6% (for social functioning) and 9.5% (for mental health) to the coefficient of determination, R^2^, for SF = 36 and 5.7% for happiness. There was no interaction between resilience and working with COVID-19 for any of the outcomes studied. Many of the coefficients of determination of the multivariate models are shown in Table 6.

## 4. Discussion

The results showed that physiotherapists with low resilience who worked with COVID-19 patients presented the lowest scores for QoL and SH. Additionally, age, sex, absence from work, receipt of PPE, receipt of host leadership, regular physical activity, and maintenance of physical activity during the pandemic were predictors of QoL and SH scores.

The measurement of QoL is an evaluation of health in the context of an individual’s perception of well-being. This global health assessment tool considers specific domains, such as physical, psychological health, social relationships, environment, mental health, financial resources, and bodily pain [69]. In our findings, the scores for all domains of the QoL evaluation (except for social functioning) were significantly lower for the low-resilience group and the group that worked with COVID-19 patients, compared with the high-resilience group and the group that did not work with COVID-19 patients. Studies have shown that healthcare workers with low resilience [70] who treat COVID-19 patients present more negative psychological symptoms, compared with those who do not work with COVID-19 patients [71,72,73], and that there is an association between negative psychological symptoms and QoL [74,75,76]. Korkmaz et al. (2020) demonstrated that physicians, nurses, and assisting healthcare staff employed in COVID-19 services showed decreased QoL due to increased anxiety levels [77]. In fact, the struggle of FHCWs also involves the maintenance of their mental health. There is a high prevalence of symptoms of post-traumatic stress disorders in this population, which can cause lead to severe negative consequences, such as lower QoL, loss of workforce, and loss of productivity [78].

As with other FHCWs, physiotherapists are exposed to intense physical and mental stress. A qualitative study was conducted in Italy during the first COVID-19 outbreak on the emotions/feelings of frontline physiotherapists. The most narrated and shared emotion experienced by all the participants was fear [79]. A high prevalence of anxiety and depressive symptoms of depression and anxiety was also detected in frontline physiotherapists in three hospitals in South Korea [80]. The causes for the increase in symptoms of depression, anxiety, and stress were multi-factorial and included work overload, exposure to the virus, experiencing prejudice, fear of contaminating others, the daily experience of death, and impact of social isolation [81].

We found no difference between the groups regarding the assessment of social functioning, for which the scores were low. Regardless of resilience level and whether or not COVID-19 patients were served, movement restrictions imposed during the pandemic to break the chain of infection limited the social interactions of healthcare workers. A study by Woon et al. (2021) on FHCWs who worked in university hospitals and healthcare facility settings with medical and allied healthcare staff (including physiotherapists) revealed that these professionals felt frustrated by the loss of a daily routine and engagement in leisure and sports activities, as well as by the separation from family and other support systems [82].

The perception of current health, compared with health a year ago, was poor among physiotherapists with low resilience who worked in COVID-19 sectors. These findings support the definition of health by the WHO as complete physical, mental, and social well-being and not merely the absence of disease or infirmity [83]. People respond to adverse life events in different ways—some respond with distress; others are more resistant and adaptable. Biological factors, cognitive features, and emotional and interpersonal hypersensitivity can determine how people react to adverse situations. People with an expressed predisposition to low stress can develop depression and, in time, different bodily disorders and illnesses [84]. However, studies suggest that resilience can protect against work-related stress [85] and has a positive effect on health. High levels of resilience may help combat adversity and stress; thus, resilience is associated with a high QoL [86]. In an observational study conducted by Hirten et al. (2021), the authors suggested the identification of employees at risk of psychological sequelae, low emotional support, and low resilience. This provides an opportunity for healthcare institutions to minimize the impact of factors associated with longitudinal stress on FHCWs [87].

Positive emotions and an increase in global life satisfaction (happiness) are characteristics of resilient people [88]. The cognitive evaluation of happiness depends on individual experiences. High SH is a protective factor against the harmful effects of negative emotions [89]. In our study, physiotherapists who showed lower scores in the evaluation of SH were those with low resilience who were working directly with COVID-19 patients. Satici et al. (2020) demonstrated that resilience was a protective factor associated with a decreased fear of COVID-19 and increased SH in the Turkish population [90]. People with high levels of happiness can better cope with difficulties, have reduced levels of stress, and have better physical health and QoL [91]. A one-year investigation of the mental health of intensivist care physicians during the pandemic in Italy concluded that nearly half of the workers often felt burnout, and levels of job satisfaction and happiness in life were not satisfactory [92].

Although resilience is considered an individual characteristic, several factors may influence this positive response, including prior psychiatric history, female sex, having young children, exposure level, working role, years of work experience, social and work support, job organization, quarantine, age, marital status, and coping styles [93]. Spilg et al. (2022) showed that younger healthcare workers and females may be more prone to low moral resilience. Indeed, many are not fully prepared to cope with the adversities of the profession and do not possess all the necessary skills [94]. Females seem to be more affected by emotional exhaustion, which is associated with low resilience. This fact seems to be evident in women who are responsible for taking care of their house and family Actions for providing and increasing resilience in FHCWs during the COVID-19 pandemic have been recommended [95,96]. Regular exercise can help enhance levels of psychological resilience, thus improving well-being [97]. This outcome was observed in our data, which showed that Brazilian physiotherapists who maintained and engaged in physical activity had better perceptions of QoL and happiness. Regular physical exercise may: increase the local expression of the brain-derived neurotrophic factor (BDNF) gene in brain areas involved in resilience, such as the hippocampus and prefrontal cortex; minimize the risk of post-traumatic stress symptoms; increase heart rate variability; and promote neuropeptide secretion [98].

Another important finding was that leadership support had a protective effect on the evaluated physiotherapists. Healthcare workers are exposed to psychosocial risks at work that can be exacerbated by exposure to contagious diseases [99]. Rangachari and Woods (2020) discussed the importance of leaders taking safe and effective actions to improve resilience and protect FHCWs from emotional distress to maintain well-being [100]. The lack of PPE has also been identified as a psychosocial risk factor that triggers fears of infection and transmission of the virus. In Jordanian FHCWs who provided care for patients with COVID-19, adequate PPE reduced anxiety and depression and increased the level of resilience of the study participants [101].

This study has a few limitations. First, given the descriptive and cross-sectional nature of the study, the results are only applicable at a given point in time. Second, the absence of a detailed assessment of the working conditions and organizations in which the participants worked could impact the results. Third, we are unaware of the total number of questionnaires distributed and the recovery rate. However, we emphasize that during the period in which the study data were collected, the number of admissions to hospitals and the number of deaths in Brazil increased, and no specific anti-viral treatment for COVID-19 and no vaccine were available.

Our findings confirm the importance of resilience for overall well-being when facing a stressor. Physiotherapists with higher resilience experienced the pandemic with higher levels of QoL and SH. This study is the first to evaluate these variables among Brazilian physiotherapists treating COVID-19 patients in hospitals. The identification of factors that determined these responses can encourage physiotherapist managers and hospital administrators to continue, or invest in, actions to improve resilience and identify the individuals at risk of developing acute and chronic stress.

## 5. Conclusions

Physiotherapists with low resilience who worked with COVID-19 patients had lower QoL and SH, compared with the others evaluated. Several factors modulated these responses: age, sex, absence from work, receipt of PPE, receipt of host leadership, regular physical activity, and maintenance of physical activity during the pandemic. Strategies that promote resilience may substantially improve the ability to deal with stressful situations. They protect against and alleviate symptoms that can compromise the physical and mental health of frontline physiotherapists.

## Figures and Tables

**Table 1 ijerph-19-08720-t001:** Demographics, socioeconomic status, and clinic dates according to low and high resilience.

Variables	Low Resilience (*N* = 145)	High Resilience (*N* = 374)	*p* Value
Age, *n* (%)			
20–30 years	35 (28.9)	115 (35.7)	0.135
31–40 years	63 (52.1)	161 (50.0)	0.933
41–50 years	20 (16.5)	39 (12.1)	0.284
51–60 years	3 (2.5)	7 (2.2)	0.857
Female, *n* (%)	127 (87.6)	327 (87.4)	0.975
Pregnancy status, *n* (%)	3 (2.1)	10 (2.7)	0.733
Marital status, *n* (%)			
Married	41 (33.9)	132 (41.0)	0.128
Divorced	7 (5.8)	13 (4.0)	0.312
Separated	1 (0.8)	1 (0.3)	0.558
Not married	59 (48.8)	150 (46.6)	0.901
Stable union	11 (9.1)	23 (7.1)	0.867
Others	2 (1.7)	3 (0.9)	0.565
Has children, *n* (%)	43 (35.5)	122 (37.7)	0.519
Family members living together, *n* %			
Seniors	25 (17.2)	73 (19.5)	0.560
Children	52 (35.9)	135 (36.1)	0.963
Death in family or close friends due to COVID-19, *n* (%)	48 (33.1)	103 (27.5)	0.214
Graduation time, *n* (%)			
˂5 years	34 (23.4)	85 (22.7)	0.854
5–10 years	54 (37.2)	119 (31.8)	0.242
11–20 years	48 (33.1)	150 (40.1)	0.141
21–30 years	9 (6.2)	20 (5.3)	0.692
Physical activity, *n* (%)			
Practice of regular physical activity *	64 (44.1)	203 (54.3)	0.038
Physical activity during the pandemic period	17 (13.3)	63 (18.5)	0.146
Medical history, *n* (%)			
Previous chronic disease	28 (19.3)	57 (15.2)	0.266
Absence from work due to other diseases	19 (13.1)	47 (12.6)	0.858
COVID-19 diagnosis	33 (22.8)	98 (26.2)	0.422
Needed hospitalization due to COVID-19	5 (3.4)	5 (1.3)	0.147
Nature of the institution where they work, *n* (%)			
Public	53 (39.0)	114 (35.7)	0.187
Private	50 (36.8)	108 (33.9)	0.216
Both	33 (24.3)	97 (30.4)	0.458
Removed from work due to, *n* (%)			
Pregnancy	2 (8.7)	5 (11.9)	0.932
A chronic disease	3 (13.0)	2 (4.8)	0.158
Adapted to work at home office	0	2 (4.8)	. ^a^
Other reasons	18 (78.3)	33 (78.6)	0.226
The hospital sector where they work, *n* (%)			
Critical care unit	110 (75.9)	264 (70.6)	0.231
Semi intensive unit	4 (2.8)	13 (3.5)	0.714
Ward	25 (17.2)	71 (19.0)	0.655
Emergency room	2 (1.4)	6 (1.6)	0.900
Supervisor occupations	2 (1.4)	11 (2.9)	0.330
Ambulatory medicine	2 (1.4)	9 (2.4)	0.506
Weekly workload, *n* (%)			
˂20 h	2 (1.4)	6 (1.6)	0.900
20–30 h	64 (44.8)	155 (41.6)	0.578
31–40 h	32 (22.4)	100 (26.8)	0.210
41–50 h	13 (9.1)	32 (8.6)	0.868
51–60 h	24 (16.8)	47 (12.6)	0.242
˃60 h	8 (5.6)	33 (8.8)	0.212
Wage/income in R$, *n* (%)			
˂1.500,00	2 (1.4)	6 (1.6)	0.900
1.500,00–3.000,00	29 (20.1)	70 (18.8)	0.733
3.000,00–5.000,00	72 (50.0)	184 (49.3)	0.925
5.000,00–7.000,00	27 (18.8)	79 (21.2)	0.533
˃7.000,00	14 (9.7)	34 (9.1)	0.829
Salary reduction during the pandemic, *n* (%)	25 (17.2)	84 (22.5)	0.190
Support for coping with the pandemic, *n* (%)			
Received personal protective equipment	136 (93.8)	358 (95.7)	0.366
Received host leadership *	83 (57.2)	263 (70.3)	0.005
Received training *	107 (73.8)	307 (82.1)	0.038

Data presented as frequency (percentage) for categorical variables. (. ^a^) This category is not used in comparisons because its column proportion is equal to zero or one. * *p* < 0.05.

**Table 2 ijerph-19-08720-t002:** Demographics, socioeconomic status, and clinic dates according to working or not with COVID-19 patients.

Variables	NO COVID (*N* = 74)	COVID (*N* = 445)	*p* Value
Age, *n* (%)			
20–30 years	22 (36.1)	128 (33.5)	0.856
31–40 years	31 (50.8)	193 (50.5)	0.816
41–50 years	6 (9.8)	53 (13.9)	0.351
51–60 years	2 (3.3)	8 (2.1)	0.588
Female, *n* (%)	68 (91.9)	386 (86.7)	0.215
Pregnancy status, *n* (%) **	7 (9.5)	6 (1.3)	0.000
Marital status, *n* (%)			
Married	18 (29.5)	155 (40.6)	0.073
Divorced	2 (3.3)	18 (4.7)	0.632
Separated	1 (1.6)	1 (0.3)	0.285
Not married	33 (54.1)	176 (46.1)	−0.415
Stable union	6 (9.8)	28 (7.3)	0.549
Others	1 (1.6)	4 (1.0)	0.688
Has children, *n* (%)	23 (37.1)	142 (37.1)	0.897
Family members living together, *n* (%)			
Seniors	13 (17.6)	85 (19.1)	0.774
Children **	40 (54.1)	147 (33.0)	0.000
Death in family or close friends due to COVID-19, *n* (%)	22 (29.7)	129 (29)	0.887
Graduation time, *n* (%)			
˂5 years	13 (17.6)	106 (23.8)	0.237
5–10 years *	14 (18.9)	159 (35.7)	0.003
11–20 years *	38 (51.4)	160 (36.0)	0.013
21–30 years *	9 (12.2)	20 (4.5)	0.017
Physical activity, *n* (%)			
Practice of regular physical activity	40 (54.1)	227 (51)	0.631
Physical activity during the pandemic *	19 (27.9)	61 (15.3)	0.013
Medical history, *n* (%)			
Previous chronic disease	12 (16.2)	73 (16.4)	0.989
Absence from work due to other diseases *	2 (2.7)	64 (14.4)	0.001
COVID-19 diagnosis	15 (20.3)	116 (26.1)	0.291
Needed hospitalization due to COVID-19	1 (1.4)	9 (2)	0.782
Nature of the institution where they work, *n* (%)			
Public	15 (26.3)	152 (38.2)	0.142
Private *	28 (49.1)	130 (32.7)	0.015
Both	14 (24.6)	116 (29.1)	0.188
Removed from work due to, *n* (%)			
Pregnancy **	5 (35.7)	2 (3.9)	0.000
A chronic disease	2 (14.3)	3 (5.9)	0.173
Adapted to work at home office	1 (7.1)	1 (2.0)	0.285
Other reasons	6 (42.9)	45 (88.2)	0.620
The hospital sector where they work, *n* (%)			
Critical care unit **	26 (35.1)	348 (78.2)	0.000
Semi intensive unit	4 (5.4)	13 (2.9)	0.293
Ward **	30 (40.5)	66 (14.8)	0.000
Emergency room	0	8 (1.8)	. ^a^
Supervisor occupations *	5 (6.8)	8 (1.8)	0.031
Ambulatory medicine **	9 (12.2)	2 (0.4)	0.000
Weekly workload, *n* (%)			
˂20 h	2 (2.7)	6 (1.4)	0.410
20–30 h	38 (52.1)	181 (40.9)	0.088
31–40 h	19 (26.0)	113 (25.5)	0.946
41–50 h	9 (12.3)	36 (8.1)	0.263
51–60 h *	3 (4.1)	68 (15.3)	0.004
˃60 h	2 (2.7)	39 (8.8)	0.060
Wage/income in R$, *n* (%)			
˂1500,00	0	8 (1.8)	. ^a^
1.500,00–3.000,00	18 (24.7)	81 (18.2)	0.223
3.000,00–5.000,00	30 (41.1)	226 (50.9)	0.104
5.000,00–7.000,00	12 (16.4)	94 (21.2)	0.338
˃7.000,00 *	13 (17.8)	35 (7.9)	0.014
Salary reduction during the pandemic, *n* (%) *	22 (29.7)	87 (19.6)	0.047
Support for coping with the pandemic, *n* (%)			
Received personal protective equipment	74 (100)	420 (94.4)	. ^a^
Received host leadership **	62 (83.8)	284 (63.8)	0.000
Received training	61 (82.4)	353 (79.3)	0.522

Data presented as frequency (percentage) for categorical variables; (. ^a^) This category is not used in comparisons because its column proportion is equal to zero or one. * *p* < 0.05; ** *p* < 0.001.

**Table 3 ijerph-19-08720-t003:** Physiotherapists’ perceptions of current health, compared with health in the previous year.

SF-36 (Question 2)			
Resilience	Low	High	*p* Value
	3 (3–4)	3 (2–4)	<0.001
Works with COVID-19 patients	No	Yes	
	3 (2–3)	3 (3–4)	<0.001

Data presented as median (interquartile range).

**Table 4 ijerph-19-08720-t004:** Prevalence of quality of life and happiness among physiotherapists who presented low and high resilience.

Resilience			
	Low	High	*p* Value
36-item Short Form Health Survey (SF-36) %			
Physical functioning	80 (60–90)	90 (75–100)	<0.001
Role-physical	50 (25–100)	75 (25–100)	<0.001
Bodily pain	51 (41–62)	62 (51–84)	<0.001
General health	60 (50–73)	72 (60–82)	<0.001
Vitality	35 (20–45)	50 (35–65)	<0.001
Social functioning	50 (37–50)	50 (37–50)	0.163
Role-emotional	33 (0–66)	100 (91–100)	<0.001
Mental health	44 (36–52)	66 (0–100)	<0.001
Subjective Happiness Scale (SHS)	4.25 (3.62–5.00)	5.50 (5.00–6.25)	<0.001
Resilience	67 (60–71)	86 (80–97)	<0.001

Data presented as median (interquartile range).

**Table 5 ijerph-19-08720-t005:** Prevalence of quality of life and happiness among physiotherapists who work or do not work with COVID-19 patients.

Works in COVID-19			
	No	Yes	*p* Value
36-item Short Form Health Survey (SF-36) %			
Physical functioning	95 (85–100)	85 (70–95)	<0.001
Role-physical	100 (75–100)	50 (25–100)	<0.001
Bodily pain	84 (62–100)	51 (41–72)	<0.001
General health	80 (61–90)	67 (55–77)	<0.001
Vitality	65 (55–80)	40 (30–55)	<0.001
Social functioning	50 (37–50)	50 (37–50)	0.638
Role-emotional	100 (91–100)	33 (0–100)	<0.001
Mental health	82 (68–92)	52 (44–64)	<0.001
Subjective Happiness Scale (SHS)	6.12 (5.25–6.75)	5.00 (4.25–5.75)	<0.001
Resilience	90 (84–95)	80 (72–88)	<0.001

Data presented as median (interquartile range).

**Table 6 ijerph-19-08720-t006:** Linear regression of quality of life and happiness associated with resilience and COVID-19.

Outcomes	Unadjusted B (SE)			Adjusted B (SE) ^#^		
	Resilience	COVID	ΔR^2^ (Resilience + COVID)	Resilience	COVID	Total R^2^
Physical functioning	8.284 (2.112) **	−6.354 (2.694) *	0.058	7.300 (2.068) **	−4.181 (2.664)	0.140
Role physical	13.152 (4.128) *	−29,667 (5.226) **	0.110	12.350 (4.102) *	−25.289 (5.285) **	0.162
Bodily pain	7.748 (2.420) **	−18.442 (3.088) **	0.119	6.934 (2.401) *	−15.553 (3.094) **	0.174
General health	8.850 (1.820) **	−7.255 (2.322) *	0.100	7.894 (1.771) **	−4.983 (2.282)	0.179
Vitality	13.879 (1.956) **	−22.417 (2.495) **	0.282	12.544 (1.903) **	−19.350 (2.452) **	0.352
Social functioning	−2.475 (1.600)	2.332 (2.041)	0.011	−2.358 (1.634)	2.515 (2.105)	0.017
Role emotional	19.102(4.487) **	−39.314(5.724) **	0.162	17.994 (4.463)	−34.649 (5.750)	0.210
Mental health	16.541 (1.159) **	−21.619 (2.243) **	0.353	14.935 (1.664) **	−18.260 (2.143)	0.448
Happiness	0.056 (0.004) **	−0.469 (0.129) **	0.403	0.053 (0.004) **	−0.363 (0.129) *	0.438

Abbreviation: B = coefficient; SE = standard error; ^#^ adjusted for age, sex, absence from work, received protective personal equipment, received host leadership, practice regular physical activity, and maintenance of physical activity during the pandemic period; ** *p* < 0.001; * *p* < 0.05; resilience is a factor (high = 1; low = 0); works in COVID-19 unit is a factor (yes = 1; no = 0).

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
