# Peer review of "Resilience Improves the Quality of Life and Subjective Happiness of Physiotherapists during the COVID-19 Pandemic"

_ijerph, 2022, doi:10.3390/ijerph19148720_

Round 1

Reviewer 1 Report

Dar Authors, 

Thank you for the effort in writing and improving the paper. I'm aware that this is resubmission. Unfortunately, I do not have access to the previous version. As such, please excuse some comments that may seem out of context.

Generally, the paper is well written, but it needs editing (for example the space between tables 1 and 2 is unnecessary.

Main main concern is in the theoretical framework:

  • the concept of happiness is highly arguable; as such, you need to critically evaluate the concept
  • being central for the study, the concept of resilience needs further development (see for example Ferreira, P., & Gomes, S. (2021). The role of resilience in reducing burnout: A study with healthcare workers during the COVID-19 pandemic. Social Sciences, 10(9), 317.), including the relation with subjective happiness and quality of life

Another concern is related with the Cronbach Alpha of one of the constructs:

  • the subjective happiness scale present a Cronbach Alpha too low, below the usual threshold; this means the items of the scale are not consistent; since this is an indicator that might influence the analysis, you need to explain this value when you present it, not in the discussion;
  • in tables 1 and 2, instead of the notation “N.S.” please include the actual value
  •  

Author Response

Reviewer 1

Comments and Suggestions for Authors

Dar Authors, 

Thank you for the effort in writing and improving the paper. I'm aware that this is resubmission. Unfortunately, I do not have access to the previous version. As such, please excuse some comments that may seem out of context.

Generally, the paper is well written, but it needs editing (for example the space between tables 1 and 2 is unnecessary.

Response: We have adjusted the space between tables 1 and 2, as suggested.

Main main concern is in the theoretical framework:

  • the concept of happiness is highly arguable; as such, you need to critically evaluate the concept
  • being central for the study, the concept of resilience needs further development (see for example Ferreira, P., & Gomes, S. (2021). The role of resilience in reducing burnout: A study with healthcare workers during the COVID-19 pandemic. Social Sciences10(9), 317.), including the relation with subjective happiness and quality of life.

Response: Thanks for the comment. We have included more information in the introduction section.

 Another concern is related with the Cronbach Alpha of one of the constructs:

  • the subjective happiness scale present a Cronbach Alpha too low, below the usual threshold; this means the items of the scale are not consistent; since this is an indicator that might influence the analysis, you need to explain this value when you present it, not in the discussion;

Response: According to the notes of the other reviewer, we redone the statistical analysis of Cronbach’s alpha of the subjective happiness scale and the corrected value is 0.816. In this way, we have corrected the material and methods and removed the explanation from the discussion, correcting this error.

  • in tables 1 and 2, instead of the notation “N.S.” please include the actual value

 Response:  We have included the actual value in tables 1 and 2.

Reviewer 2 Report

I cannot agree with a study in which one of the two main explanatory variables is measured with an unreliable tool. 

I would advise the authors:

to remove it from the analyses, or

to check the data for unreliably completed surveys, or

to check with the questionnaire key to ensure that all statements were coded correctly. 

Respectfully yours, 

Author Response

Reviewer 2

Comments and Suggestions for Authors

I cannot agree with a study in which one of the two main explanatory variables is measured with an unreliable tool. 

I would advise the authors:

to remove it from the analyses, or

to check the data for unreliably completed surveys, or

to check with the questionnaire key to ensure that all statements were coded correctly. 

Respectfully yours, 

Response: Thanks for the reviewer’s advice. The questions 3 and 4 of the subjective happiness scale ask the same question in a different ways, what results in a negative correlation, and for this reason the value of Cronbach’s alpha was low. The question coding has been reversed, so now the corrected value of Cronbach’s alpha corresponds to 0.816. We are grateful for the pertinent comment and consequent contribution as it greatly improved the statistical analysis.

Round 2

Reviewer 2 Report

I am glad that the error was found and corrected. I am surprised that the change in the coding of the results of one question is not reflected in the change in the results in the Tables presenting the data analysis (Tables 4-6). The values of the variable happiness should change after recoding, and consequently the relationship indicators too. It is imperative that this be checked and possibly corrected.

Author Response

I am glad that the error was found and corrected. I am surprised that the change in the coding of the results of one question is not reflected in the change in the results in the Tables presenting the data analysis (Tables 4-6). The values of the variable happiness should change after recoding, and consequently the relationship indicators too. It is imperative that this be checked and possibly corrected.

 Response: We agree with the notes made by the reviewer. We did the recoding of encoded the variables, recalculated the descriptive statistics, and redone the multiple regression. We greatly appreciate the considerations as right after this correction the results were more consistent with higher R2. The values in tables 4-6 have been updated.

We have also observed that no longer find the interaction between resilience and working with COVID-19. In this way, we have removed this information from the text in the results section (regression analysis) and in the discussion section. We have included the following sentence in section 3.4 (regression analysis): “For none of the outcomes studied there was interaction between resilience and work with COVID-19 patients.”

This manuscript is a resubmission of an earlier submission. The following is a list of the peer review reports and author responses from that submission.

Round 1

Reviewer 1 Report

One of used measurement - The Subjective Happiness Scale (SHS) is not valid. Cronbach’s alpha for this scale was 0.401. It should be excluded from the analysis.

Reviewer 2 Report

Dear Authors,

Thanks for the improvements made in the manuscript.

You have reviewed the manuscript accordingly.